# Structural basis of agonist specificity of α$_{1A}$-adrenergic receptor

Minfei Su[1,5], Jinan Wang[2,5], Guoqing Xiang ⓘ [3,4], Hung Nguyen Do ⓘ [2], Joshua Levitz ⓘ [3,4], Yinglong Miao ⓘ [2] ✉ & Xin-Yun Huang ⓘ [1] ✉

α$_1$-adrenergic receptors (α$_1$-ARs) play critical roles in the cardiovascular and nervous systems where they regulate blood pressure, cognition, and metabolism. However, the lack of specific agonists for all α$_1$ subtypes has limited our understanding of the physiological roles of different α$_1$-AR subtypes, and led to the stagnancy in agonist-based drug development for these receptors. Here we report cryo-EM structures of α$_{1A}$-AR in complex with heterotrimeric G-proteins and either the endogenous common agonist epinephrine or the α$_{1A}$-AR-specific synthetic agonist A61603. These structures provide molecular insights into the mechanisms underlying the discrimination between α$_{1A}$-AR and α$_{1B}$-AR by A61603. Guided by the structures and corresponding molecular dynamics simulations, we engineer α$_{1A}$-AR mutants that are not responsive to A61603, and α$_{1B}$-AR mutants that can be potently activated by A61603. Together, these findings advance our understanding of the agonist specificity for α$_1$-ARs at the molecular level, opening the possibility of rational design of subtype-specific agonists.

Epinephrine and norepinephrine are neurotransmitters of the sympathetic nervous system, and hormones secreted by the adrenal medulla[1,2]. They function through nine distinct human adrenergic receptor (AR) subtypes: α$_1$-type (α$_{1A}$, α$_{1B}$, α$_{1D}$), α$_2$-type (α$_{2A}$, α$_{2B}$, α$_{2C}$,) and β-type (β$_1$, β$_2$, and β$_3$)[3,4]. Due to the lack of selective pharmacological agonists for α$_1$-ARs, the therapeutic potential of α$_1$-ARs has been largely unexplored[5–9]. However, mouse genetic studies with individual gene deletions have demonstrated distinct but overlapping physiological functions of α$_1$-AR subtypes in the regulation of blood pressure, cardiac hypertrophy, vascular smooth muscle contraction, neurotransmission, learning and memory, and metabolism[10–17]. For example, the α$_{1A}$-AR subtype is a vasopressor expressed in resistance arteries and is required for normal arterial blood pressure regulation[11]. In addition, α$_{1B}$-AR in human coronary endothelial cells mediates vasodilation[18,19].

α$_1$-ARs couple to the Gq family of G-proteins, leading to the stimulation of phospholipase C-β to cleave phosphatidylinositol-4,5-bisphosphate into inositol-1,4,5-trisphosphate and 2-diacylglycerol[20]. The former promotes the release of Ca$^{2+}$ from intracellular stores, while the latter activates protein kinase C[21,22]. While some pharmacological compounds exist that can target α$_1$-ARs, there are only two structurally related α$_{1A}$-specific agonists, and no α$_{1B}$-AR and α$_{1D}$-AR selective agonists have been reported[17]. A61603 is a high affinity, selective α$_{1A}$-AR agonist which shows almost no activity at α$_{1B}$-AR and α$_{1D}$-AR[7,23]. Furthermore, there are no structures available for the active states of α$_1$-ARs. In this paper, we use cryo-EM to determine the structures of α$_{1A}$-AR/Gq signaling complexes with epinephrine or A61603. These structures reveal the molecular basis for the binding specificity of A61603 for α$_{1A}$-AR, and the different conformations of epinephrine in interacting with α-ARs versus β-ARs. Gaussian accelerated molecular dynamics (GaMD) simulations and functional studies provide further insights into the mechanisms of specificity, ultimately enabling the validation of key sites that determine the ability of A61603 to specifically activate α$_{1A}$-AR but not α$_{1B}$-AR[24,25].

[1]Department of Physiology and Biophysics, Weill Cornell Medical College of Cornell University, New York, NY 10065, USA. [2]Center for Computational Biology and Department of Molecular Biosciences, University of Kansas, Lawrence, KS 66047, USA. [3]Department of Biochemistry, Weill Cornell Medical College of Cornell University, New York, NY 10065, USA. [4]Department of Psychiatry, Weill Cornell Medical College of Cornell University, New York, NY 10065, USA. [5]These authors contributed equally: Minfei Su, Jinan Wang. ✉e-mail: miao@ku.edu; xyhuang@med.cornell.edu

## Results

### Cryo-EM structures of A61603–α₁ₐ-AR–Gq and epinephrine–α₁ₐ-AR–Gq signaling complexes

To understand the agonist specificity of $\alpha_1$-AR, we solved cryo-EM structures of human $\alpha_{1A}$-AR bound to A61603 (a synthetic specific agonist for $\alpha_{1A}$-AR) at 2.6 Å or epinephrine (an endogenous agonist for all ARs) at 3.0 Å, both in complex with its cognate signaling Gq heterotrimer (mini-G$\alpha_q$G$\beta_1$G$\gamma_2$) (Fig. 1a, b, Supplementary Figs. 1–3, and Supplementary Table 1). Comparisons of these structures should provide insights into subtype-specific agonist binding. Overall, the structures of the $\alpha_{1A}$-AR–Gq complex in the presence of epinephrine or A61603 are similar (Fig. 1a, b). However, there are local conformational differences, especially in the ligand-binding pockets (see below). While some of the interacting residues are common to both ligands, epinephrine, and A61603 each make a unique set of interactions in the orthosteric ligand-binding pocket (see below).

Since the A61603–$\alpha_{1A}$-AR–Gq and epinephrine–$\alpha_{1A}$-AR–Gq structures reveal the first active state conformation of $\alpha_1$-AR family

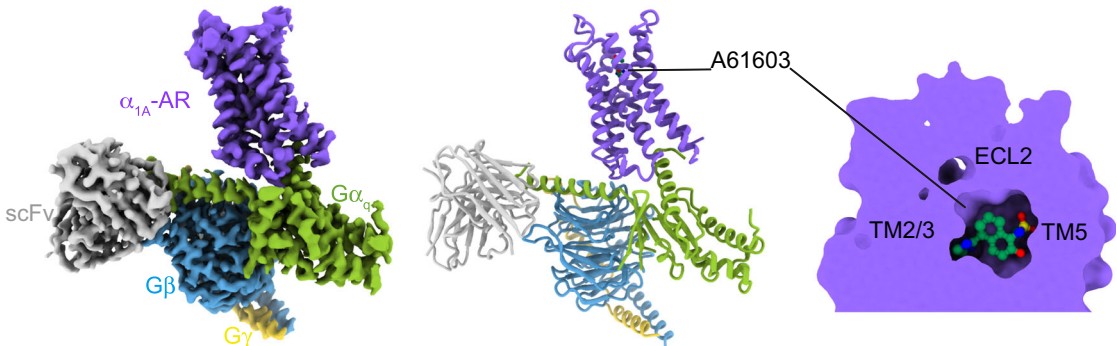

**a**   Selective agonist (A61603)–α₁ₐ-AR–Gq complex

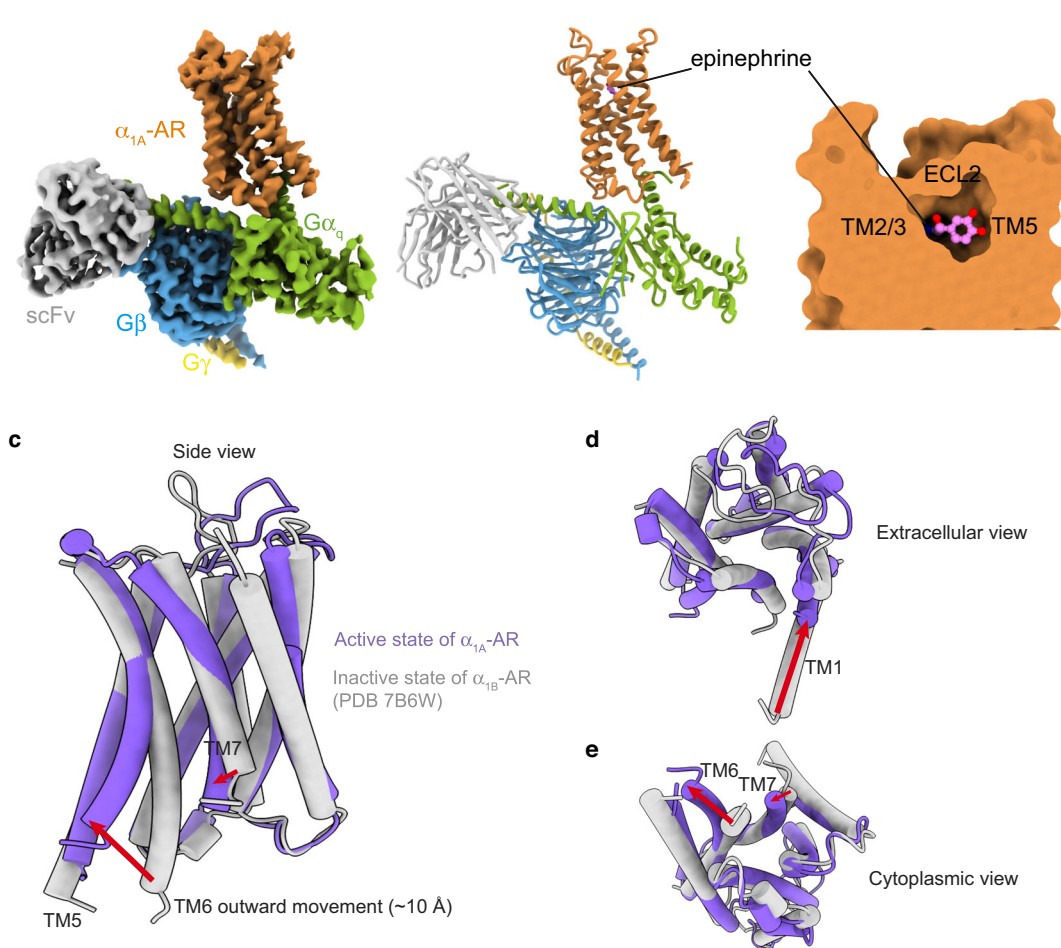

**b**   Non-selective endogenous agonist (epinephrine)–α₁ₐ-AR–Gq complex

**Fig. 1 | Cryo-EM structures of the complexes of A61603–α₁ₐ-AR–Gq and epinephrine–α₁ₐ-AR–Gq. a** The density map, the model, and the ligand-binding pocket of A61603–α₁ₐ-AR–Gq are shown. **b** The density map, the model, and the ligand-binding pocket of epinephrine–α₁ₐ-AR–Gq are shown. A61603-bound α₁ₐ-AR is colored in purple. Epinephrine-bound α₁ₐ-AR is colored orange. Gα_q in green. Gβ in blue. Gγ in yellow. **c–e** Different views of the inactive state α₁ᵦ-AR (gray; PDB 7B6W) and the active state of α₁ₐ-AR in complex with Gq (this work).

receptors, we first investigated the structural basis of activation. For the α$_1$-AR family, there is only one recently reported structure: the X-ray crystal structure of the inactive state of α$_{1B}$-AR bound with the inverse agonist cyclazosin (PDB: 7B6W)[26]. We thus compared our active state structures of α$_{1A}$-AR with this inactive state structure of α$_{1B}$-AR. This comparison revealed characteristic conformational changes associated with class A GPCR activation (Fig. 1c–e, and Supplementary Fig. 4)[27]. Since the structures of α$_{1A}$-AR bound with A61603 and epinephrine were similar, we focused our comparison on the A61603–α$_{1A}$-AR–Gq complex. The overall root-mean-square deviation between the structures of the active α$_{1A}$-AR and inactive α$_{1B}$-AR is 1.6 Å over 198 Cα atoms. The largest structural changes upon activation occur on the cytoplasmic side of α$_1$-AR (Fig. 1c–e), with an outward rotation of TM6 by ~10 Å (measured at the Cα of A$^{6.33}$) (The superscript is the Ballesteros-Weinstein numbering[28]) (Fig. 1c, e), and an inward ~5 Å movement of TM7 (measured at the Cα of Y$^{7.53}$) (Fig. 1c, e). In addition to these TM conformational changes, rearrangements of side chains of certain residues are observed as part of the α$_1$-AR activation process. Just below the orthosteric ligand-pocket, the rotameric change of W$^{6.48}$ (within the CWxP motif) denotes the opening of TM6 in class A GPCRs for G-protein engagement[27] (Supplementary Fig. 4a). A layer below the CWxP motif, the rotation of F$^{5.47}$ causes the change of L$^{5.51}$, accompanied by the translational movement of V$^{6.45}$, F$^{6.44}$ (part of

the PIF motif), and W$^{6.48}$ to form new contacts (Supplementary Fig. 4b). Moving closer to the G-protein-interacting site below the PIF motif, the highly conserved NPxxY motif at the cytoplasmic end of TM7 is another key micro-switch of GPCR activation[27] (Supplementary Fig. 4c). TM7 rotates around the NPxxY motif. This moves Y$^{7.53}$ toward the position that was occupied by TM6 in the inactive structure (Supplementary Fig. 4c). Among the G-protein-interacting residues, the rearrangement of side chains in the highly conserved D(E)/RY motif in TM3 is critical for GPCR activation[27] (Supplementary Fig. 4d). Therefore, α$_1$-ARs undergo conformational changes propagating from the orthosteric ligand-binding site to the G-protein-interacting site during its activation.

## Different conformations of epinephrine bound to α-ARs and β-ARs

Epinephrine is a chiral endogenous full agonist common for all ARs. In the complex of epinephrine–α$_{1A}$-AR–Gq, the para-hydroxyl group of the catechol ring of epinephrine forms H-bond with the hydroxyl side-chain of S188$^{5.43}$ (Fig. 2a–c and Supplementary Fig. 5). The protonated amine of epinephrine forms a salt bridge with residue D106$^{3.32}$ (Fig. 2a–c and Supplementary Fig. 5). Previous experiments have shown that this interaction is critical for both affinity and efficacy[29]. Y316$^{7.42}$ and W313$^{7.39}$ stabilize this salt bridge through a hydrogen bond

**a**  Chemical structure of (−)-epinephrine

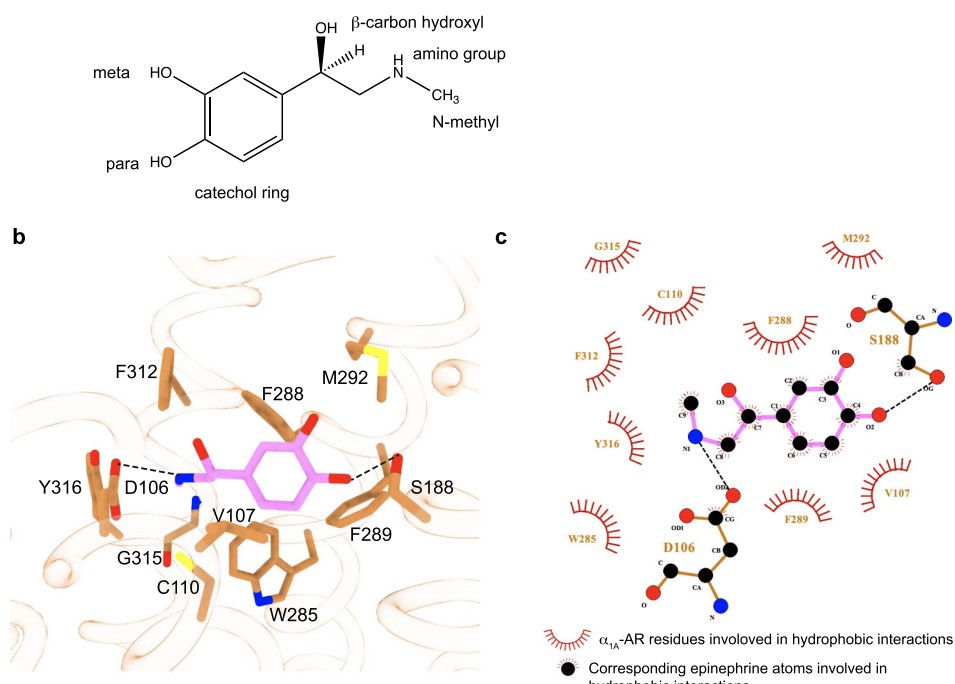

**d**  Different conformations of epinephrine in complex with α-ARs and with β-ARs

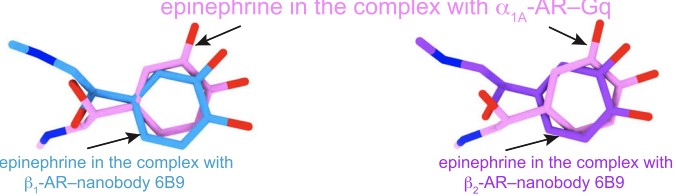

**Fig. 2 | Interactions between epinephrine and α$_{1A}$-AR. a** Chemical structure of epinephrine. **b** Schematic diagram of the epinephrine-binding pocket of α$_{1A}$-AR from the cryo-EM structure is shown. **c** Ligplot diagrammatic representation of interactions between epinephrine and α$_{1A}$-AR. **d** Comparisons of the conformations of epinephrine bound with α$_{1A}$-AR, β$_1$-AR, and β$_2$-AR.

between D106[3.32] and Y316[7.42], and π–π interactions between Y316[7.42] and W313[7.39] (Fig. 2b, c). The catechol ring forms π–π interactions with F288[6.51] and F289[6.52] (Fig. 2c). F312[7.38] acts as a lid covering the catechol ring of epinephrine from the extracellular side (Fig. 2b). Furthermore, epinephrine adopted a binding conformation similar to the cryo-EM structure in the GaMD simulations (Supplementary Fig. 5a), with mostly <2 Å root-mean-square derivation (RMSD) (Supplementary Fig. 5b). The ligand maintained stable interactions with residues D106[3.32] and S188[5.43] in $\alpha_{1A}$-AR at $3.82 \pm 0.33$ Å and $3.10 \pm 0.33$ Å, respectively (Supplementary Fig. 5c, d).

The conformation of epinephrine in the complex with $\alpha_{1A}$-AR is different from that in the previously reported complexes with β-ARs (Fig. 2d and Supplementary Fig. 6). Currently there are no structures available for epinephrine in complex with any β-ARs and a G-protein. However, there are two structures of epinephrine bound to $\beta_1$-AR or $\beta_2$-AR with a nanobody 6B9 that stabilizes β-ARs in an active state[30,31]. In the complex of epinephrine–$\beta_1$-AR–nanobody 6B9 and of epinephrine–$\beta_2$-AR–nanobody 6B9[30,31], the β-carbon hydroxyl and the N-methyl group face opposite directions when compared with the conformation in complex with $\alpha_{1A}$-AR. The β-carbon hydroxyl faces the intracellular side, and the N-methyl group points toward TM3 when epinephrine was complexed with β-ARs (Fig. 2d, Supplementary Fig. 6). On the other hand, the β-carbon hydroxyl group faces the extracellular side, and the N-methyl group points to Y316[7.42] on TM7, when epinephrine was in the complex with $\alpha_{1A}$-AR (Fig. 2b, Supplementary Fig. 6). These two different epinephrine conformations result from the rotation around the chemical bond linking the catechol ring and the β-carbon of epinephrine (Supplementary Fig. 6). In β-ARs, N363[7.38] (in $\beta_1$-AR) and N312[7.38] (in $\beta_2$-AR) interacted through hydrogen bonds with the β-carbon hydroxyl and the N-methyl group of epinephrine. In $\alpha_{1A}$-AR, it is F312[7.38] at the equivalent position, and its side-chain did not form hydrogen bonds with epinephrine (Supplementary Fig. 6). The different conformations of epinephrine, when bound with β-ARs and α-ARs, might explain the higher affinity of epinephrine for β-ARs than α-ARs[32]. Together these data show that epinephrine adopts different conformations in order to interact with α and β families of ARs.

## Specific interaction between A61603 and $\alpha_{1A}$-AR

Since A61603 is a selective agonist for $\alpha_{1A}$-AR[23], we asked what residues mediated this specific interaction. In the complex of A61603–$\alpha_{1A}$-AR–Gq, S188[5.43] forms multiple interactions with A61603 (Fig. 3a–c). The hydroxyl group of S188[5.43] forms H-bonds with the hydroxy attached to the tetrahydronaphthalene group, the amide, and the oxygen atom of the methanesilfonamide group of A61603 (Fig. 3b, c). In addition, the side-chain of D106[3.32] forms a salt bridge with the imidazoline group of A61603 (Fig. 3b, c), and the backbone amide hydrogen of A189[5.44] interacts with the oxygen atom of the methane-silfonamide group of A61603 (Fig. 3b, c). There are also extensive hydrophobic interactions between A61603 and $\alpha_{1A}$-AR (Fig. 3c). These include V107[3.33], C110[3.36], I178[45.52], W285[6.48], F288[6.51], F289[6.52], M292[6.55], F312[7.38], G315[7.41] and Y316[7.42] (Fig. 3c). Since this is the first structure of A61603 bound to any proteins, we performed GaMD simulations to validate the ligand pose (Fig. 3d–g and Supplementary Fig. 7). In these simulations, the conformation of A61603 is similar to that observed in the cryo-EM structure (Fig. 3d, e). The imidazoline group of A61603 and residue D106[3.32] in $\alpha_{1A}$-AR maintained the stable salt-bridge interaction during the GaMD simulations (Fig. 3f). Residue S188[5.43] in $\alpha_{1A}$-AR formed hydrogen bonds with different atoms (e.g., N, O, and $O_2$) in the methanesulfonamide group of A61603 in the GaMD simulations (Fig. 3g and Supplementary Fig. 7). Together, these data reveal the molecular interactions between A61603 and $\alpha_{1A}$-AR.

## Molecular basis for the discrimination between $\alpha_{1A}$-AR and $\alpha_{1B}$-AR by A61603

To further probe the structural basis for the specificity of A61603 for $\alpha_{1A}$-AR, we compared the ligand-binding pockets of all three $\alpha_1$-AR subtypes (Supplementary Fig. 8). Among the ligand-binding pocket residues, $\alpha_{1A}$-AR and $\alpha_{1B}$-AR have almost identical composition with the exception of three residues: V185[5.40] in $\alpha_{1A}$-AR but A204[5.40] in $\alpha_{1B}$-AR, A189[5.44] in $\alpha_{1A}$-AR and S208[5.44] in $\alpha_{1B}$-AR, and M292[6.55] in $\alpha_{1A}$-AR but L314[6.55] in $\alpha_{1B}$-AR (Fig. 4a and Supplementary Fig. 8b).

In the A61603–$\alpha_{1A}$-AR–Gq structure, these three residues form a hydrophobic surface surrounding the methanesulfonamide group of A61603 (Fig. 4b). To understand why A61603 could not bind to $\alpha_{1B}$-AR, we modeled a complex of $\alpha_{1B}$-AR and A61603 (Fig. 4c). Based on our cryo-EM structure of the active state of $\alpha_{1A}$-AR with A61603 and the structure of the inactive state of $\alpha_{1B}$-AR, the modeled A61603–$\alpha_{1B}$-AR complex showed a steric clash between the methanesulfonamide group of A61603 and L314[6.55] (Fig. 4c). In $\alpha_{1D}$-AR, the corresponding residue is also a Leu residue (Supplementary Fig. 8b). Together, these observations suggest a molecular basis for the specific binding of A61603 to $\alpha_{1A}$-AR, but not $\alpha_{1B}$-AR and $\alpha_{1D}$-AR.

Based on insights from our structures and molecular modeling, we attempted to engineer $\alpha_{1A}$-AR variants that would lose the response to A61603, and $\alpha_{1B}$-AR mutants that would gain the response to A61603. These loss-of-function and gain-of-function studies should provide evidence for the molecular basis of A61603-binding specificity. Based on the above data, we focused on three residues V185[5.40], A189[5.44,] and M292[6.55]. We generated point mutants of $\alpha_{1A}$-AR(V185A), $\alpha_{1A}$-AR(A189S), and $\alpha_{1A}$-AR(M292L) by mutating these residues to the corresponding residues in $\alpha_{1B}$-AR. Since these residues are involved in hydrophobic interactions with A61603 (Fig. 3), we also made a mutant with all three residues mutated $\alpha_{1A}$-AR(V185A, A189S, M292L). Since $\alpha_{1A}$-AR is coupled to Gq, we used calcium responses as a functional readout (Fig. 5). As a control, wild-type $\alpha_{1A}$-AR produced robust dose-dependent $Ca^{2+}$ responses to A61603 with an $EC_{50}$ of ~1 nM with a similar amplitude at saturating doses compared to saturating epinephrine (Fig. 5a, c). The triple mutant $\alpha_{1A}$-AR(V185A, A189S, M292L) reduced the apparent affinity of the A61603 response by more than 1000-fold (Fig. 5b, c). This triple mutant was still functional in response to epinephrine (Fig. 5b). The single-point mutants of V185A and M292L also decreased the $\alpha_{1A}$-AR response to A61603, while the A189S mutant showed wild-type-like responses (Fig. 5c).

In parallel, GaMD simulations were performed on $\alpha_{1A}$-AR(V185A), $\alpha_{1A}$-AR(A189S), $\alpha_{1A}$-AR(M292L), and $\alpha_{1A}$-AR(V185A, A189S, M292L) (Supplementary Fig. 9). During the GaMD simulations, RMSDs of A61603 in $\alpha_{1A}$-AR(V185A), $\alpha_{1A}$-AR(M292L), and $\alpha_{1A}$-AR(V185A, A189S, M292L) relative to the starting cryo-EM structure were $4.62 \pm 1.68$ Å, $3.21 \pm 1.05$ Å, and $3.99 \pm 1.13$ Å, respectively (Supplementary Fig. 9a, g, j). Hence, $\alpha_{1A}$-AR(V185A), $\alpha_{1A}$-AR(M292L), and $\alpha_{1A}$-AR(V185A, A189S, M292L) reduced the binding of A61603 to the receptor (Supplementary Fig. 9a, g, j). On the other hand, A61603 in $\alpha_{1A}$-AR(A189S) exhibited a smaller RMSD with lower fluctuations in the GaMD simulations (Supplementary Fig. 9d). The distances between the imidazoline group of A61603 and residue D106[3.32] in $\alpha_{1A}$-AR(V185A), $\alpha_{1A}$-AR(A189S), $\alpha_{1A}$-AR(M292L), and $\alpha_{1A}$-AR(V185A, A189S, M292L) were $4.25 \pm 0.98$ Å, $3.43 \pm 0.47$ Å, $4.13 \pm 1.33$ Å, and $4.38 \pm 1.27$ Å, respectively (Supplementary Fig. 9b, e, h, k). Therefore, the interaction between the imidazoline group of A61603 and residue D106[3.32] was disrupted in $\alpha_{1A}$-AR(V185A), $\alpha_{1A}$-AR(M292L), and $\alpha_{1A}$-AR(V185A, A189S, M292L), but still maintained in $\alpha_{1A}$-AR(A189S), during the simulations (Supplementary Fig. 9b, e, h, k). This is consistent with our functional data where V185A and M292L produced a large impairment of A61603 responses while A189S maintained WT-like properties. For further information, we performed molecular mechanics/generalized Born surface area (MM/GBSA)

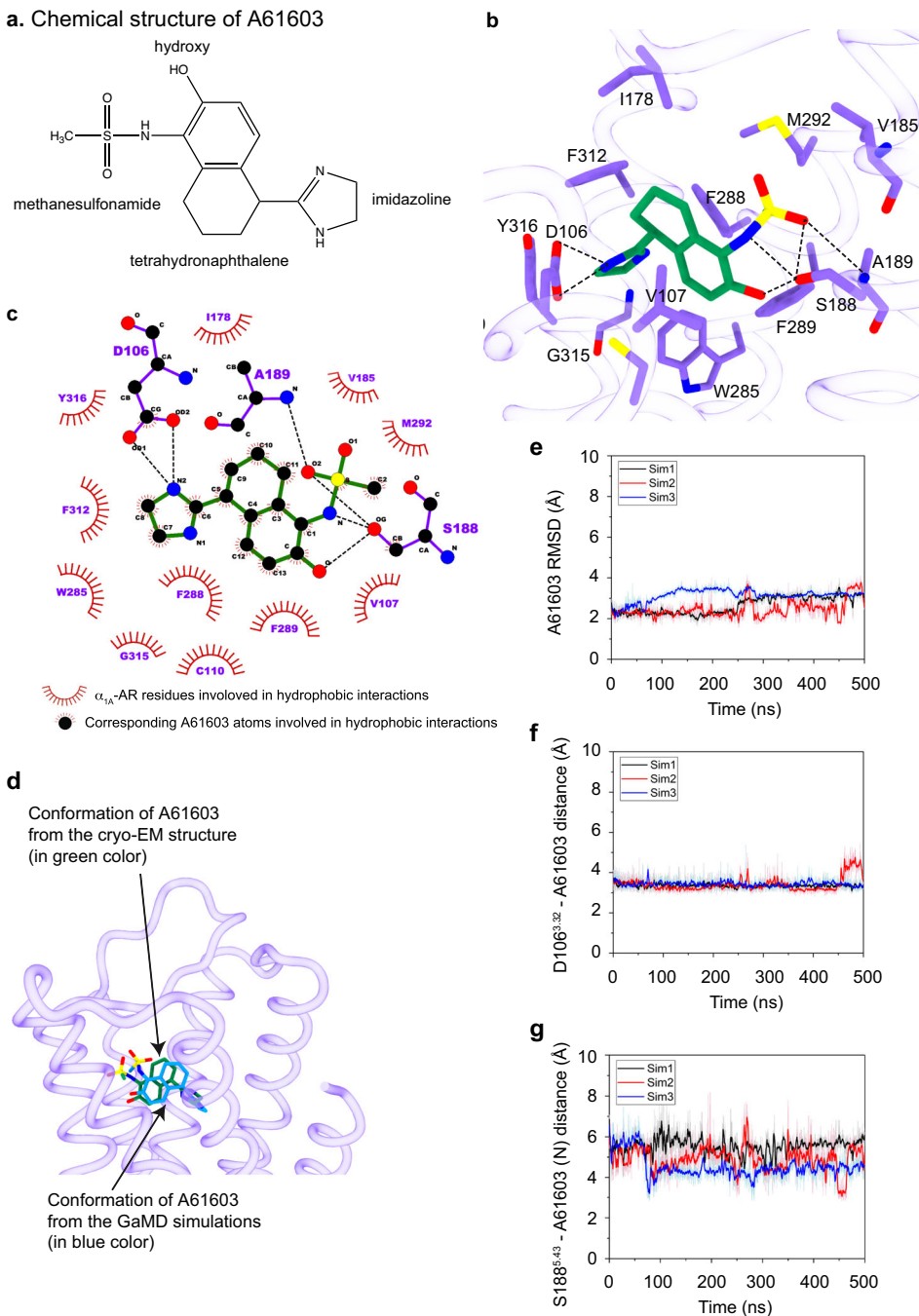

**Fig. 3 | Interactions between A61603 and α₁ₐ-AR. a** Chemical structure of A61603. **b** Schematic diagram of the A61603-binding pocket of α₁ₐ-AR from the cryo-EM structure is shown. **c** Ligplot diagrammatic representation of interactions between A61603 and α₁ₐ-AR. **d** Comparison of the binding poses of A61603 in the cryo-EM structure (in green color) and from GaMD simulations (in blue color). **e** The A61603 binding free energy calculations using GaMD trajectories[33–35]. Due to the RMSD by comparing the conformations of A61603 in the cryo-EM structure and from GaMD simulations. **f, g** Time courses of the distances between A61603 and specific residues of α₁ₐ-AR were calculated from the GaMD simulations: the CG atom of D106$^{3.32}$ and the N2 atom of A61603 (**f**), and the OG atom of S188$^{5.43}$ and the N atom of A61603 (**g**). Three independent 500 ns simulations are shown.

binding free energy calculations using GaMD trajectories[33–35]. Due to the inherent inaccuracy of entropy calculations, we focused on comparing only the enthalpy values (Supplementary Table 2). The A61603-binding free energy was less favorable in the α₁ₐ-AR mutants, i.e., −22.18 ± 0.73 kcal/mol for α₁ₐ-AR(V185A) and −11.52 ± 2.47 kcal/mol for α₁ₐ-AR(V185A, A189S, M292L), compared to −26.73 ± 0.22 kcal/mol for the wild-type α₁ₐ-AR. α₁ₐ-AR(A189S) exhibited a similar binding free energy of A61603 (−26.54 ± 1.89 kcal/mol) as the wild-type α₁ₐ-AR. However, an exception was observed in α₁ₐ-AR(M292L), where the binding free energy was −27.88 ± 1.53 kcal/mol. This likely resulted from

the exclusion of the entropy and MM/GBSA free energy calculations of GaMD simulation frames without the energetic reweighting[36]. In general, the systems with a stronger functional response to A61603-binding exhibited more favorable binding free energy compared to those with weaker or no functional responses. Therefore, the GaMD simulation findings correlated well with the calcium response data.

Conversely, we attempted to convert α₁ᵦ-AR from non-responsive to responsive to A61603. This type of exercise is usually difficult given that many residues are typically involved in ligand interactions. However, based on our above data, most of the agonist interacting residues

**a**

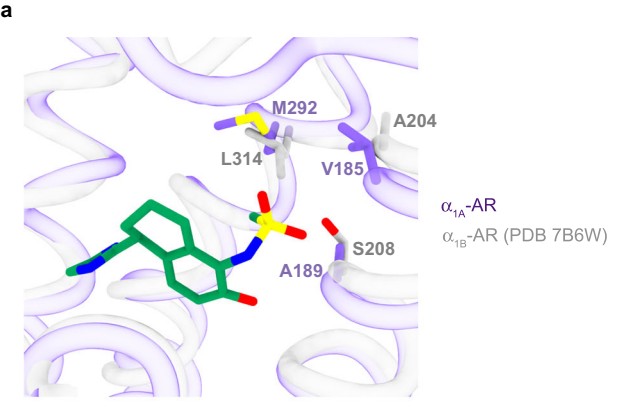

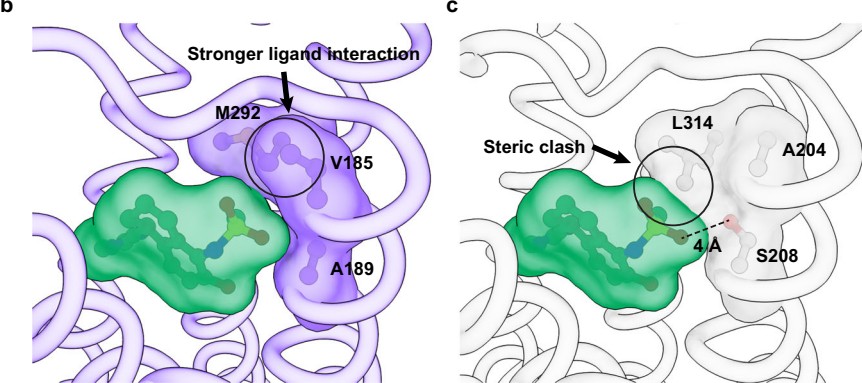

**Fig. 4 | Participation of V185, A189, and M292 of $\alpha_{1A}$-AR in interacting with A61603. a** Comparison of the ligand-binding pockets of $\alpha_{1A}$-AR and $\alpha_{1B}$-AR. **b** Hydrophobic interaction between the methanesulfonamide group of A61603 and the three residues V185, A189, and M292 of $\alpha_{1A}$-AR. **c** Steric clash between the docked A61603 and $\alpha_{1B}$-AR.

are the same between $\alpha_{1A}$-AR and $\alpha_{1B}$-AR, except for the three residues (Supplementary Fig. 8), suggesting that it may be possible to replace the three distinct residues in $\alpha_{1B}$-AR by those in $\alpha_{1A}$-AR. We thus generated the corresponding triple mutant $\alpha_{1B}$-AR(A204V, S208A, L314M). Wild-type $\alpha_{1B}$-AR did not respond to A61603 at concentrations up to 10 μM, while it was responsive to epinephrine (Fig. 5d). Remarkably, the triple mutant $\alpha_{1B}$-AR(A204V, S208A, L314M) responded to A61603 in a dose-dependent manner with an EC$_{50}$ around 10 nM (Fig. 5e, f). The single-point mutants $\alpha_{1B}$-AR(A204V), $\alpha_{1B}$-AR(S208A) or $\alpha_{1B}$-AR(L314M) alone did not respond to A61603 (Fig. 5f), indicating that more of a large-scale reshaping of the orthosteric site than can be achieved by a single mutation is needed to rescue the binding. In parallel, we have built a homology model of the active $\alpha_{1B}$-AR using the cryo-EM structure of A61603-bound $\alpha_{1A}$-AR as the template and conducted GaMD simulations on the wild-type $\alpha_{1B}$-AR, $\alpha_{1B}$-AR(S208A), and $\alpha_{1B}$-AR(A204V, S208A,L314M). In wild-type $\alpha_{1B}$-AR and $\alpha_{1B}$-AR(S208A), A61603 exhibited large RMSDs relative to the starting conformation, with reduced interactions between A61603 and $\alpha_{1B}$-AR (Supplementary Fig. 10). In contrast, A61603 in $\alpha_{1B}$-AR(A204V, S208A, L314M) exhibited low RMSD at ~2.5 Å, forming stable interactions with residues D125$^{3.32}$ and S207$^{5.43}$ in the receptor (Supplementary Fig. 10). $\alpha_{1B}$-AR(A204V, S208A, L314M) showed the strongest binding affinity (-28.21 ± 2.80 kcal/mol), followed by $\alpha_{1B}$-AR(S208A) (-25.81 ± 2.67 kcal/mol) and wild-type $\alpha_{1B}$-AR (-11.52 ± 2.47 kcal/mol) (Supplementary Table 2).

We also compared the root-mean-square fluctuations (RMSF) of A61603 in different receptor systems. In this context, larger RMSF values indicate weaker binding of A61603. The calculated RMSF values of A61603 in $\alpha_{1A}$-AR (WT), $\alpha_{1A}$-AR(V185A), $\alpha_{1A}$-AR(A189S), $\alpha_{1A}$-AR(M292L), and $\alpha_{1A}$-AR(V185A, A189S, M292L)

were 1.46 ± 0.10 Å, 2.36 ± 0.52 Å, 1.10 ± 0.088 Å, 1.74 ± 0.39 Å, and 2.58 ± 0.33 Å, respectively (Supplementary Table 2). $\alpha_{1A}$-AR(A189S) (with a functional response to A61603) exhibited a similar RMSF value as wild-type $\alpha_{1A}$-AR, while the other $\alpha_{1A}$-AR mutants (with impaired functional responses to A61603) showed higher A61603 fluctuations. The RMSF values of A61603 in $\alpha_{1B}$-AR(A204V, S208A, L314M), $\alpha_{1B}$-AR(S208A) and wild-type $\alpha_{1B}$-AR were 0.98 ± 0.21 Å, 3.55 ± 1.97 Å, and 4.28 ± 0.30 Å, respectively (Supplementary Table 2). Additionally, we calculated the 2D free energy profiles of A61603 RMSD and the distance between the CG atom of D106$^{3.32}$ and N$_2$ atom of A61603, as well as the distance between the OG atom of S188$^{5.43}$ and O2 atom of A61603 in both the $\alpha_{1A}$-AR and $\alpha_{1B}$-AR systems (Supplementary Fig. 11). In wild-type $\alpha_{1A}$-AR and $\alpha_{1A}$-AR(A189S), one low-energy state of A61603 with a narrow conformational space was sampled. Conversely, the other mutant $\alpha_{1A}$-ARs sampled larger conformational space with at least two distinct low-energy states. Among the $\alpha_{1B}$-AR systems, $\alpha_{1B}$-AR(A204V, S208A, L314M) sampled only one low-energy state, while $\alpha_{1B}$-AR(WT) and $\alpha_{1B}$-AR(S208A) sampled at least two low-energy states with much larger conformational space (Supplementary Fig. 12). These findings suggest that the systems with functional responses to A61603 binding are sampling only one stable A61603-binding pose, whereas the systems without functional responses to A61603 could explore larger conformational space, indicating reduced stability of A61603. Together, our studies present molecular insights into the mechanism underlying the discrimination between $\alpha_{1A}$-AR and $\alpha_{1B}$-AR by A61603. The observed differences in selectivity are consistent with a model that is affinity-based, as opposed to efficacy-based.

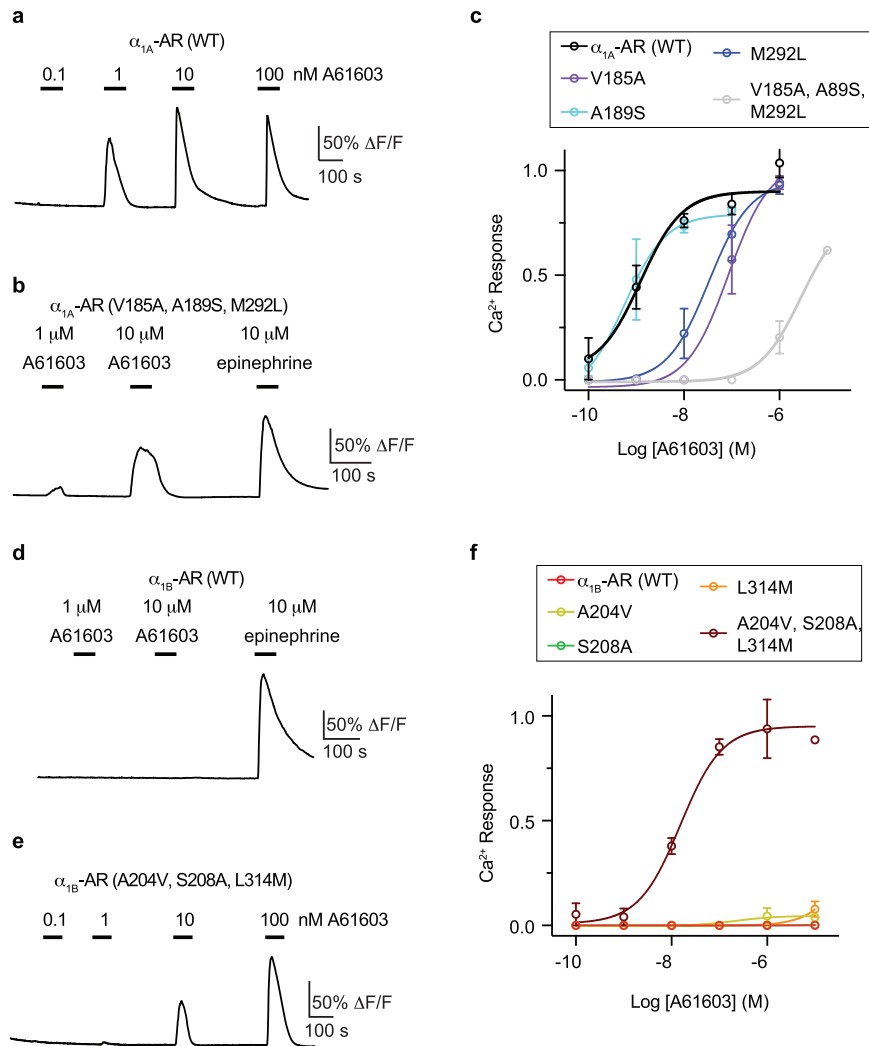

**Fig. 5 | Functional studies of mutant $\alpha_{1A}$-AR and $\alpha_{1B}$-AR. a** Representative Ca$^{2+}$ response traces of cells expressing wild-type $\alpha_{1A}$-AR in response to different concentrations of A61603. **b** Representative Ca$^{2+}$ response traces of cells expressing the triple mutant $\alpha_{1A}$-AR(V185A, A189S, M292L) in response to different concentrations of A61603 and a saturating concentration of epinephrine. **c** Dose–response curves for different $\alpha_{1A}$-AR constructs in response to A61603. $n = 3$ independent experiments. Data are presented as mean ± SD. **d** Representative Ca$^{2+}$ response traces of cells expressing wild-type $\alpha_{1B}$-AR in response to different concentrations of A61603 and a saturating concentration of epinephrine. **e** Representative Ca$^{2+}$ response traces of cells expressing the triple mutant $\alpha_{1B}$-AR(A204V, S208A, L314M) in response to different concentrations of A61603. **f** Dose–response curves for different $\alpha_{1B}$-AR constructs in response to A61603. Calcium imaging were analyzed using ImageJ and NIS-Elements Advance Research 5.2.6 software. Each single cell was manually selected as region of interest and the fluorescence intensity was normalized to the baseline before agonist application. An average trace (60-200 cells per recording) was used to quantify maximum fluorescence amplitude after drug application. $n = 3$ independent experiments. Data are presented as mean ± SD. Dose–response curve was done using nonlinear fit in GraphPad Prism.

## Discussion

$\alpha_1$-ARs are one of the three major subfamilies of ARs. Due to the lack of specific agonists, the physiological functions of $\alpha_1$-ARs are still not completely revealed, and their therapeutic potentials in treating cardiovascular, neurological, neuropsychiatric, and inflammatory disorders are not fully exploited. Here we have investigated the structural basis of activation and agonist interactions with $\alpha_1$-AR. Our data shows that M292$^{6.55}$ and V185$^{5.40}$ of $\alpha_{1A}$-AR define the specificity of A61603 for $\alpha_{1A}$-AR. Hydrophobic M292$^{6.55}$ in $\alpha_{1A}$-AR corresponds to hydrophobic L$^{6.55}$ in $\alpha_{1B}$-AR and $\alpha_{1D}$-AR, while aromatic Y$^{6.55}$ in $\alpha_2$-AR and polar N$^{6.55}$ in $\beta$-ARs (Supplementary Fig. 8b). Hydrophobic V185$^{5.40}$ in $\alpha_{1A}$-AR corresponds to A$^{5.40}$ in $\alpha_{1B}$-AR, $\alpha_{1D}$-AR, $\beta_1$-AR and $\beta_2$-AR, I$^{5.40}$ in $\alpha_{2B}$-AR and $\alpha_{2C}$-AR, and V$^{5.40}$ in $\alpha_{2A}$-AR and $\beta_3$-AR (Supplementary Fig. 8b). Given that these residues form one hydrophobic surface (Fig. 4), this hydrophobic pocket should be explored for the design of AR subtype-specific ligands.

Comparisons of the two agonists (epinephrine and A61603) revealed that they share some conserved interactions while also displaying certain unique interactions (Supplementary Fig. 13). Epinephrine is a monoamine that contains an amino group linked to an aromatic ring by a two-carbon chain (Fig. 2a). In $\alpha_{1A}$-AR, $\beta_1$-AR and $\beta_2$-AR, the aromatic ring forms $\pi$–$\pi$ interactions to F$^{6.51}$ and F$^{6.52}$ (Fig. 2b, c). The positively charged amino group forms a salt bridge to D$^{3.32}$ (Fig. 2c). Although A61603 is not a monoamine, it uses the imidazoline group to form the same salt bridge with the conserved D$^{3.32}$ (Fig. 3c and Supplementary Fig. 13). Its polycyclic aromatic ring consists of a benzene fused to cyclohexane (Fig. 3a). The benzene of A61603, as the catechol ring of epinephrine, forms $\pi$–$\pi$ interactions to the same F$^{6.51}$ and F$^{6.52}$ (Fig. 3b, c, and Supplementary Fig. 13). Furthermore, the para-hydroxyl group of epinephrine and the hydroxyl group of A61603 interact with the same S$^{5.43}$ (Figs. 2, 3 and Supplementary Fig. 13).

As discussed above, the molecular basis for the receptor specificity lies in the hydrophobic interaction of the methenesulfonamide group of A61603 with the hydrophobic surface formed by $M^{6.55}$ and $V^{5.40}$, whereas the meta-hydroxyl group of epinephrine occupies this space (Figs. 2 and 3). The differences in selectivity are consistent with a model that is affinity-based, as opposed to efficacy-based. Together our results provide insights into the design and development of selective agonists targeting individual $\alpha_1$-AR subtypes, and polypharmacological agonists tailored for multiple related receptors.

## Methods

### Expression and purification of $\alpha_{1A}$-AR, $G_q$, and scFv16

The human $\alpha_{1A}$-AR construct used in the cryo-EM study was optimized by truncation of N-terminal residues 1-14, third intracellular loop (ICL3) residues 228-255, and C-terminal residues 351-466. A hemagglutinin (HA) signal peptide, a FLAG tag, and a T4 lysozyme were fused to its N-terminus, and a PreScission protease cleavage site, an eGFP, and an 8xHis tag were added to its C-terminus. This construct was subcloned into pFastBac1 vector for expression in *Spodoptera frugiperda* Sf9 insect cells. Sf9 cells were grown to 2 to 3 million cells per ml in ESF 921 protein-free medium (Expression Systems) before 50 ml of baculoviruses were added for infection. 72 h later, cells were harvested by centrifugation, flash frozen in liquid nitrogen, and stored at −80 °C until use. For $\alpha_{1A}$-AR purification, thawed cell pellets were lysed with 1% n-Dodecyl-β-D-Maltopyranoside (DDM, Anatrace) in a buffer containing 25 mM HEPES (pH 7.0), 350 mM NaCl, 100 µM A61603 (Cayman Chemicals) or epinephrine (MP Biomedicals), supplemented with protease inhibitors (0.5 µM PMSF, 2 µg/ml leupeptin, 0.8 µM aprotinin and 2 µM Pepstatin A, Goldbio) at 4 °C. Insoluble fractions were removed by ultracentrifugation at $142,000 \times g$ for 1 h at 4 °C. The supernatant was then incubated with house-made GFP nanobody beads for at least 2 h, and washed with a buffer containing 25 mM HEPES (pH 7.0), 350 mM NaCl, 0.05% DDM, and 100 µM A61603 or epinephrine before PreScission protease was added to elute $\alpha_{1A}$-AR from GFP nanobody beads overnight at 4 °C. Eluted $\alpha_{1A}$-AR was further purified by size-exclusion chromatography using a Superdex 200 Increase 10/300 column (GE Healthcare) equilibrated with a buffer containing 25 mM HEPES (pH 7.0), 150 mM NaCl, 0.02% Lauryl Maltose Neopentyl Glycol (LMNG, Anatrace), and 100 µM A61603 or epinephrine. Peak fractions were pooled and concentrated for complex assembly.

The expression constructs of heterotrimeric mini-$G\alpha_{qiN}$–$G_{\beta1}$–$G_{\gamma2}$ and scFv16 were kindly provided by Dr. Bryan L Roth (University of North Caroline at Chapel Hill)[37]. For the expression of heterotrimeric mini-$G\alpha_{qiN}$–$G_{\beta1}$–$G_{\gamma2}$, a single virus that encodes all three subunits, including mini-$G\alpha_{qiN}$, $G_{\beta1}$, and $G_{\gamma2}$, with a 6xHis tag fused to the N-terminus of $G_{\beta1}$, was used to infect Sf9 cells. 48 h post infection, cells were harvested by centrifugation, flash frozen in liquid nitrogen, and stored at −80 °C. Cell pellets were thawed in a lysis buffer containing 20 mM HEPES (pH 7.5), 100 mM NaCl, 2 mM β-mercaptoethanol, 1 mM $MgCl_2$, 0.1 mM GDP, 0.2% (v/v) Triton X-100, supplemented with protease inhibitors (0.5 µM PMSF, 2 µg/ml leupeptin, 0.8 µM aprotinin and 2 µM Pepstatin A) at 4 °C and insoluble fractions were removed by centrifugation at $142,000 \times g$ for 1 h. Supernatant was collected and incubated with HisPur Ni-NTA Resin (Thermo Fisher Scientific) for 2 h at 4 °C. Resin was then washed with 20 mM imidazole in the lysis buffer, and eluted with 250 mM imidazole in the lysis buffer. Eluted heterotrimeric G-protein was further purified by size-exclusion chromatography using a Superdex 200 Increase 10/300 column (GE Healthcare) equilibrated with a buffer containing 20 mM HEPES (pH 7.5), 100 mM NaCl, 0.02% LMNG, 0.1 mM TCEP, 1 mM $MgCl_2$, and 10 µM GDP. Peak fractions were pooled and concentrated for complex assembly.

scFv16 was expressed and purified from Sf9 cells as a secreted protein[37]. 72 h after infection, Sf9 cell culture media expressing scFv16 was collected and PH balanced to pH 8.0 by addition of Tris buffer. 10 mM calcium chloride was then added to quench the chelating agents with stirring at room temperature for 1 h. The precipitate was removed by centrifugation at $8000 \times g$ for 30 min. The supernatant was incubated with HisPur Ni-NTA Resin with stirring for 4 h at 4 °C. The Ni-NTA resin was then collected with centrifugation at 800 g and washed with 20 mM imidazole in a buffer containing 20 mM HEPES (pH 7.5) and 100 mM NaCl before 200 mM imidazole in the same buffer was used to elute the protein. The C-terminal 8xHis tag was removed by the treatment of PreScission protease overnight at 4 °C. The protein was then further purified by size-exclusion chromatography using a Superdex 200 Increase 10/300 column (GE Healthcare) equilibrated with a buffer containing 20 mM HEPES (pH 7.5) and 100 mM NaCl. Peak fractions were pooled and concentrated for complex assembly.

### Protein complex assembly and purification

To assemble the $\alpha_{1A}$-AR–$G_q$–scFv16 complex, $\alpha_{1A}$-AR, and heterotrimeric $G_q$ were mixed at a 1:1.5 molar ratio and incubated for 30 min at room temperature followed by the treatment of 0.4 U Apyrase (New England Biolabs) for another 30 min at room temperature. A 1.5 molar access of scFv16 was then added to the mixture and incubated overnight at 4 °C. The next day, the mixture was centrifuged at $16,000 \times g$ for 10 min to remove any precipitants. The supernatant was then loaded onto a Superdex 200 Increase 10/300 column equilibrated with 10 mM HEPES pH 7.0, 100 mM NaCl, 0.1 mM TCEP, 0.02% LMNG, and 30 µM A61603 or epinephrine. The elution fractions from a single peak containing pure $\alpha_{1A}$-AR–$G_q$–scFv16 complex concentrated to 2.5 mg/ml and used directly for making cryo-EM grids.

### Cryo-EM data collection

3.5 µL of $\alpha_{1A}$-AR–$G_q$–scFv16 complex at a concentration of 2.5 mg/ml was applied to glow-discharged 400 mesh gold Quantifoil R1.2/1.3 holey carbon grids (Quantifoil Micro Tools) and vitrified using a Vitrobot Mark IV (Thermo Fisher Scientific/FEI) at 22 °C and 100% humidity. Micrographs were collected on a 300 kV Titan Krios electron microscope (Thermo Fisher Scientific/FEI) with a Gatan K3 direct electron detector (Gatan, Inc.) in super-resolution mode at a nominal ×81,000 and ×64,000 magnification for the A61603 and epinephrine-bound complexes, respectively. For the A61603-bound complex, 7,773 movies in the defocus range of −1.0 to −1.8 µm were recorded with a total accumulated dose of 50 e⁻/Å². For the epinephrine-bound complex, 10,486 movies in the defocus range of −1.0 to −1.8 µm were recorded with a total accumulated dose of 52 e⁻/Å².

### Image processing, 3D reconstructions, modeling, and refinement

Super-resolution movies were aligned, two-times Fourier cropped, and dose-weighted using MotionCor2 implemented in Relion 4.0-beta[38–44]. The effects of contrast-transfer function were estimated with CTFFIND v4.1.8[45]. For the A61603 dataset, Relion 2D template-based auto-picking was used to pick particles and the resulting particle stacks were two-times Fourier cropped and processed through 2D classification in CryoSparc v3.3.1 to remove false positives, receptor alone, or G-proteins alone particles[46] (Supplementary Fig. 1). A stack of 1,497,830 intact complex particles were then subjected to 3D classification in Relion. For the epinephrine dataset, both Relion 2D and 3D template-based auto-picking was used to pick particles, and the resulting particle stacks from both 2D and 3D template-based auto-picking were separately two-times Fourier cropped and processed through 2D classification in CryoSparc v3.3.1 to remove false positives, receptor alone, or G-proteins alone particles[46] (Supplementary Fig. 2). After Relion 3D classification, the intact complex particles stacks from 2D and 3D template-based auto-picking were selected and combined, and duplicate particles were removed. The resulting stack of 880,531

intact complex particles were then subjected to another round of 3D classification in Relion. Stacks of 773,300 (A61603) and 782,191 (epinephrine) particles that went to high resolution were selected after 3D classification and further polished by Bayesian polishing in Relion. 3D variability analysis in cryoSPARC revealed the twisting of the $\alpha_{1A}$-AR transmembrane region around a vertical axis, perpendicular to the membrane in both A61603 and epinephrine datasets. Subsets of 360,489 (A61603) and 219,834 (epinephrine) particles obtained by 3D variability analysis that yielded higher resolution in the transmembrane region were selected and non-uniform refinement of these particles in CryoSparc yielded maps with 2.6 Å-resolution (A61603) and 3.0 Å-resolution (epinephrine) using the 0.143 Fourier Shell Correlation criterion. These consensus stacks were also subjected to focused refinement of $\alpha_{1A}$-AR and $G\alpha_q$ region in CryoSparc. The focused refinement map showed significant improvement in the $\alpha_{1A}$-AR region compared to the consensus map. Models were built starting from AlphaFold predicted structure of inactive $\alpha_{1A}$-AR (AF-P35348-F1) and scFv16 bound Gq heterotrimer (PDB code 6WHA) in Coot v0.9.1, which combined with consensus and focused refinement maps, were used to generate composite maps in Phenix dev-4694[47,48]. The resulting maps were super-sampled in Coot to 0.856 Å per pixel (A61603) and 0.8608 Å (epinephrine) with 320-voxel boxes. The A61603 and epinephrine bound $\alpha_{1A}$-AR−$G_q$−scFv16 models were real-space refined against the composite maps in Phenix, and a work/free half-map pair was used to ensure against over-fitting (Supplementary Figs. 1 and 2). The binding poses of A61603 and epinephrine were initially determined by density maps as well as their geometry restraints and were further supported by MD simulations[49]. All model statistics were validated using MolProbity[50].

## Gaussian accelerated molecular dynamics (GaMD)
GaMD is an enhanced sampling method that works by adding a harmonic boost potential to reduce the system energy barriers[24,25]. When the system potential $V(\vec{r})$ is lower than a reference energy $E$, the modified potential $V^*(\vec{r})$ of the system is calculated as:

$$V^*(\vec{r}) = V(\vec{r}) + \Delta V(\vec{r})$$

$$\Delta V(\vec{r}) = \begin{cases} \frac{1}{2}k\left(E - V(\vec{r})\right)^2, & V(\vec{r}) < E \\ 0, & V(\vec{r}) \geq E, \end{cases} \quad (1)$$

where $k$ is the harmonic force constant. The two adjustable parameters $E$ and $k$ are automatically determined on three enhanced sampling principles. First, for any two arbitrary potential values $v_1(\vec{r})$ and $v_2(\vec{r})$ found on the original energy surface, if $V_1(\vec{r}) < V_2(\vec{r})$, $\Delta V$ should be a monotonic function that does not change the relative order of the biased potential values; i.e., $V_1^*(\vec{r}) < V_2^*(\vec{r})$. Second, if $V_1(\vec{r}) < V_2(\vec{r})$, the potential difference observed on the smoothened energy surface should be smaller than that of the original; i.e., $V_2^*(\vec{r}) - V_1^*(\vec{r}) < V_2(\vec{r}) - V_1(\vec{r})$. By combining the first two criteria and plugging in the formula of $V^*(\vec{r})$ and $\Delta V$, we obtain

$$V_{max} \leq E \leq V_{min} + \frac{1}{k}, \quad (2)$$

Where $V_{min}$ and $V_{max}$ are the system minimum and maximum potential energies. To ensure that Eq. 2 is valid, $k$ has to satisfy: $k \leq 1/(V_{max} - V_{min})$. Let us define: $k = k_0 \cdot 1/(V_{max} - V_{min})$, then $0 < k_0 \leq 1$. Third, the standard deviation (SD) of $\Delta V$ needs to be small enough (i.e., narrow distribution) to ensure accurate reweighting using cumulant expansion to the second order: $\sigma_{\Delta V} = k(E - V_{avg})\sigma_V \leq \sigma_0$, where $V_{avg}$ and $\sigma_V$ are the average and SD of $\Delta V$ with $\sigma_0$ as a user-specified upper limit (e.g., $10k_BT$) for

accurate reweighting. When $E$ is set to the lower bound $E = V_{max}$ according to Eq. 2, $k_0$ can be calculated as

$$k_0 = \min(1.0, k_0') = \min\left(1.0, \frac{\sigma_0}{\sigma_V} \cdot \frac{V_{max} - V_{min}}{V_{max} - V_{avg}}\right), \quad (3)$$

Alternatively, when the threshold energy $E$ is set to its upper bound $E = V_{min} + 1/k$, $k_0$ is set to:

$$k_0 = k_0'' \equiv \left(1 - \frac{\sigma_0}{\sigma_V}\right) \cdot \frac{V_{max} - V_{min}}{V_{avg} - V_{min}}, \quad (4)$$

If $k_0''$ is calculated between 0 and 1. Otherwise, $k_0$ is calculated using Eq. 3.

## System setup and simulation analysis
The epinephrine−$\alpha_{1A}$-AR−$G_q$ and A61603−$\alpha_{1A}$-AR−$G_q$ cryo-EM structures were used for setting up simulation systems to explore the stability of the complex. The missing resides in the Helix 8 of $\alpha_{1A}$-AR (KKAFQNVLR, residue number 334-342) were added by SWISS-MODEL[51]. The initial models of the A61603-bound mutants of $\alpha_{1A}$-AR(V185A), $\alpha_{1A}$-AR(A189S), $\alpha_{1A}$-AR(M292L), and $\alpha_{1A}$-AR(V185A, A189S, M292L) were built by VMD based on the A61603−$\alpha_{1A}$-AR−$G_q$ cryo-EM structure. All six simulation systems were prepared with the CHARMM-GUI web server for using the membrane protein input generator[52]. The receptor was inserted into a palmitoyl-oleoyl-phosphatidyl-choline (POPC) bilayer. All chain termini were capped with neutral patches (acetyl and methylamide). All the disulfide bonds in the complexes that were resolved in the cryo-EM structures were maintained in the simulations. The systems were solvated in 0.15 M NaCl solution at temperature 310 K. The AMBER ff14SB and AMBER LIPID 21 parameter sets were used for the receptor and lipids[53,54]. The GAFF2 parameters with RESP charges were used for epinephrine and A61603[55]. For each of the complex systems, initial energy minimization, thermalization, and 20 ns cMD equilibration were performed before GaMD simulations. A cutoff distance of 9 Å was used for the van der Waals and short-range electrostatic interactions and the long-range electrostatic interactions were computed with the particle-mesh Ewald summation method[56]. A 2-fs integration time step was used for all MD simulations. The SHAKE algorithm was applied to all hydrogen-containing bonds. With all other atoms fixed, the lipid tails were energy minimized for 5000 steps using the conjugate gradient algorithm and melted with a constant number, volume, and temperature (NVT) run for 0.5 ns at 310 K. The six systems were further equilibrated using a constant number, pressure, and temperature (NPT) run at 1 atm and 310 K for 1 ns with 1.0 kcal/(mol Å$^2$) harmonic position restraints applied to the protein and ligand atoms. Then, conventional MD simulations were performed on each system for 20 ns at 1 atm pressure and 310 K with a constant ratio constraint applied on the lipid bilayer in the X-Y plane. The GaMD module implemented in the GPU version of AMBER22 was then applied to perform the simulations[24,57]. GaMD simulations included an 8-ns short cMD simulation used to collect the potential statistics for calculating GaMD acceleration parameters, and a 48-ns equilibration after adding the boost potential. Finally, three independent 500-ns GaMD simulations with randomized initial atomic velocities were performed for the epinephrine−$\alpha_{1A}$-AR−$G_q$, A61603−$\alpha_{1A}$-AR−$G_q$, and A61603-bound mutant $\alpha_{1A}$-AR−$G_q$ complexes. The average and SD of the system potential energies were calculated every 800,000 steps (1.6 ns). All GaMD simulations were run at the "dual-boost" level by setting the reference energy to the lower bound. One boost potential was applied to the dihedral energetic term and the other to the total potential energetic term. The upper limit of the boost potential SD, $\sigma_0$ was set to 6.0 kcal/mol for both the dihedral and the total potential energetic terms.

CPPTRAJ was used to analyze the GaMD simulations[58]. Root-mean-square fluctuations (RMSFs) were calculated for the agonist A61603, averaged over three independent GaMD simulations. Interactions between epinephrine and the receptor were measured by the distance between the CG atom of D106 and N1 atom of epinephrine, and the OG atom of S188 and $O_2$ atom of epinephrine. Interactions between A61603 and the receptor were measured by the distance between the CG atom of D106 and the N2 atom of A61603, the N atom of A189 and the O2 atom of A61603, the OG atom S188 and the N atom of A61603, the OG atom of S188 and the O atom of A61603, and the OG atom of S188 and the O2 atom of A61603. Time courses of these reaction coordinates obtained from the GaMD simulation were plotted in Fig. 3, Supplementary Figs. 5, 7, 9, and 10.

## Calcium imaging

HEK293T cells were purchased from ATCC (CRL-11268; CRL-1573), authenticated by Bio-Synthesis, Inc. and routinely tested negative for mycoplasma. Cells were maintained in DMEM supplemented with 10% fetal bovine serum (FBS) in 5% $CO_2$ at 37 °C in humidified incubator. HEK293T cells were seeded on poly-L-lysine coated glass coverslips (18 mm) one day before transfection. GCaMP 6 f (0.2 μg per well) and $\alpha_{1A}$-AR plasmid (0.5 μg per well) were used to transfect cells using Lipofectamine 2000 in 12-well plates. Cells were protected with 10 μM Prazosin 6–8 hours after transfection. Calcium imaging experiments were performed 24 hours after transfection. Cells were placed on an inverted Nikon Eclipse Ti2-E microscope equipped with an Andor Zyla 5.5 sCMOS camera, Lumencor SOLA-SE II light engine, and CFI Nikon Plan Apo Lambda ×20 with 0.75 NA. GCaMP6f was excited using 488 nm SOLA LED light. Cells were continuously perfused with extra-cellular (EX) solution (containing (in mM): 135 NaCl, 5.4 KCl, 10 HEPES, 2 $CaCl_2$, 1 $MgCl_2$, pH = 7.4) utilizing a gravity-driven perfusion system at room temperature. Drugs (A61603 or epinephrine) were dissolved in EX solution. Calcium imaging was analyzed using ImageJ 1.52a and NIS-Elements Advance Research 5.2.6. Each single cell was manually selected as region of interest and the fluorescence intensity was normalized to the baseline before agonist application. An averaged trace (60–200 cells per recording) was used to quantify maximum fluorescence amplitude after drug application. Dose–response curve was done using nonlinear fit in GraphPad Prism.

## Quantification and statistical analysis

In Fig. 5, the $Ca^{2+}$ assays were repeated three times, and the data are represented as mean ± SD of the three independent experiments. The analysis was done using the log(agonist) vs. response function of Prism 9 (GraphPad) as indicated in the figure legends. Cryo-EM data collection and refinement statistics are listed in Supplementary Table 1.

## Reporting summary

Further information on research design is available in the Nature Portfolio Reporting Summary linked to this article.

## Data availability

The data that support this study are available from the corresponding authors upon request. The cryo-EM reconstructions of the A61603–$\alpha_{1A}$-AR–$G_q$–scFv16 complex and the epinephrine–$\alpha_{1A}$-AR–$G_q$–scFv16 complex have been deposited in the Election Microscopy Data Bank (EMDB) under accession codes EMD-41267 (the A61603–$\alpha_{1A}$-AR–$G_q$–scFv16 complex) and EMD-41268 (the epinephrine–$\alpha_{1A}$-AR–$G_q$–scFv16 complex), respectively. The corresponding atomic models have been deposited in the Protein Data Bank (PDB) under ID codes 8THK (the A61603–$\alpha_{1A}$-AR–$G_q$–scFv16 complex) and 8THL (the epinephrine–$\alpha_{1A}$-AR–$G_q$–scFv16 complex), respectively. Publicly available PDB entries used in this study are available under the accession codes 7B6W and 6WHA. The source data underlying Fig. 5c

and 5f are provided as a Source Data file. Trajectories for molecular dynamics simulations can be found in the Figshare repository [https://doi.org/10.6084/m9.figshare.23664450.v1] and [https://doi.org/10.6084/m9.figshare.23664270.v1]. Source data are provided with this paper.

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

## Acknowledgements
We thank members of our research groups for helpful discussion and comments on the manuscript, the Laboratory for BioMolecular Structure (L.B.M.S.) staff, and the New York Structural Biology Center staff for the help with the cryo-EM data acquisition. This work was supported by NIH grants GM138676 (X.Y.H.) and GM132572 (Y.M.), as well as AHA grant 23CDA1049353 (https://doi.org/10.58275/AHA.23CDA1049353.pc.gr.168015) (M.S.). This work used supercomputing resources with allocation award TG-MCB180049 through the Extreme Science and Engineering Discovery Environment (XSEDE) and project M2874 through the National Energy Research Scientific Computing Center (NERSC), and the Research Computing Cluster and BigJay Cluster resource funded through NSF Grant MRI-2117449 at the University of Kansas. L.B.M.S. is supported by the DOE Office of Biological and Environmental Research (KP1607011). Some of this work was performed at the Simons Electron Microscopy Center and National Resource for Automated Molecular

Microscopy located at the New York Structural Biology Center, supported by grants from the Simons Foundation (SF349247), NYSTAR, the Agouron Institute (F00316), and the NIH (GM103310, OD019994).

## Author contributions

M.S. expressed and purified α$_{1A}$-AR, mini-GαqiN–Gβ1–Gγ2, and the protein complexes, made cryo-EM grids, performed cryo-EM screening, data collection, EM density map determination, and model building. J.W. and H.N.D. performed GaMD simulations under the supervision of Y.M. G.X. performed the calcium imaging under the supervision of J.L. X.Y.H. supervised the project, interpreted data, and wrote the manuscript. All authors contributed to the final version of the manuscript.

## Competing interests

The authors declare no competing interests.
