## [Peer Review File · Nature Communications]

Structural Basis of Agonist Specificity of α 1A-Adrenergic ReceptorReviewers' Comments:

Reviewer #1:

Remarks to the Author:

The manuscript by Su et al. reports novel structural insights for the α_1A Adrenoceptor. The two cryo-EM structures add a missing piece to the structural knowledge about the adrenoceptor family and nicely explain subtype selectivity of two closely related receptors on a molecular level. While I'm not an expert for cryo-EM, the data seems to be reliable and sound, but I cannot fully judge on this part. The topic of the manuscript is of interest for a broad readership and deserves publication. However, some issues regarding chemical terms the mutational data and the MD simulations have to be addressed before.

1) The authors should carefully check the manuscript for the right chemical terms/nomenclature. Epinephrine has only one configuration, but can have different conformations, which should not be mixed up (2nd paragraph of the results, Fig. S6, ...). A rotation of a single bond doesn't result in different stereoisomers (1133-135). The ligand A6160 doesn't contain an imidazole ring, but an imidazoline ring (e.g. Figure 3, 1156, ...).

2) The mutational studies for understanding the selectivity of A61603 are convincing, but it could benefit from a more detailed discussion. The results for α_1A A189S and α_1B S208A are quite surprising, what's the explanation for this? The authors just briefly touch this topic. The figure 4C shows the α_1B receptor (pdb entry 7B6W). In this structure the α_1B receptor is bound to an inverse agonist. I wonder whether and how this influences the result. In any case, some modeling of the α_1B receptor would strengthen the manuscript (not necessary to model all mutations!).

3) Why do the authors use GAMD and not classical MD simulations? I wonder, if calculated free energies would support the findings of the simulated mutants.

4) I would suggest to reorganize Figure S9 to directly compare systems with the same mutation. The plots show quite some divergent behavior for the single runs. Running simulations in triplicates is state-of-the-art, but if they behave so differently another run (or two) might be necessary. I miss a brief explanation what is happening during the simulations, which might help two address the issue mentioned above.

Reviewer #2:

Remarks to the Author:

In this study, Su and colleagues have solved two CryoEM structures of the active-state α_1A adrenergic receptor in complex with a 'mini-Gq' transducer and either the endogenous agonist, epinephrine (3 A) or the selective agonist, A61603 (2.6 A). This finding is important because, although α_1 adrenergic receptor-targeting therapeutics have been available for many years, they are largely non-selective and thus many opportunities remain for more development of more selective agents for a range of unmet medical needs. By and large, the study is well performed and the application of molecular dynamics and targeted mutagenesis to validate and provide new insights into determinants of agonist selectivity between the related α_1A and α_1B subtypes is a useful contribution to the field. I have some comments for consideration by the authors to strengthen the manuscript and their findings.

First, in terms of writing style, I feel that there are a few instances of either ambiguity or over-statement. For instance, in the Introduction, the authors comment on a lack of 'selective pharmacological agents for α_1 -ARs' as a general statement, and also cite a relatively old reference (in fact, a number of references are surprising old or not appropriate – more below). Do they mean lack of selective antagonists, selective agonists, or both? Clearly, this paper is focusing on the latter and, as I indicated, there are already many α_1R -targeting medicines on the market. This needs to be

made clearer.

Second, it is possible that I missed this, but I find it surprising that the authors have not actually characterised the binding affinity of their agonists in their own hands on the constructs used for structural determination. At a minimum, this needs to be done, and it is not sufficient to simply quote ref. 24, Dalal & Grujic, StatPearls (2022). Indeed, I would like to draw the authors' attention to a recent publication by Proudman and Baker, 2021, Pharmacol. Res. Persp., 9(4): e00799, where they carefully compared the pharmacology of 62 different alpha AR agonists, including A61603, confirming its high selectivity, but speculating that this likely driven by preferential binding rather than preferential signaling efficacy. This is important to include, and relates to the third section of the current MS ("Different configurations of epinephrine....etc."), where the authors identify interactions critical for '...both potency and efficacy'....This statement is incorrect; I assume the authors mean '...both affinity and efficacy...'. Thus, in light of the above, it is necessary to have a comparative estimate of the affinity of each of the agonists used, in addition to the functional data presented by the authors.

Third, there are a number of instances where I feel that the authors make too much of the different configurations of epi that can be exploited to produce alpha- or beta-AR selective agonists; either be more explicit about some potential modifications that may be worth pursuing in the future, or else tone this down.

The most interesting part of the study in my opinion is the molecular discrimination between $\alpha 1A$ and $\alpha 1b$ selectivity (many of the early findings on the active states are not particularly surprising when compared to existing active state Class A receptors). The targeted mutagenesis supports that data but, as I note, only in function. I feel that this could also be supported by some carefully selected mutants to validate in binding assays as well.

Finally, the Discussion is rather repetitive of the results. I think that, if the data were supplemented with additional binding experiments, it would be more useful for the field to Discuss the implications of affinity-based versus efficacy-based selectivity of such agonists.

List of Manuscript Changes

We thank the reviewers very much for the helpful comments on our manuscript.

Reviewer #1:

“The manuscript by Su et al. reports novel structural insights for the $\alpha 1A$ Adrenoceptor. The two cryo-EM structures add a missing piece to the structural knowledge about the adrenoceptor family and nicely explain subtype selectivity of two closely related receptors on a molecular level. While I’m not an expert for cryo-EM, the data seems to be reliable and sound, but I cannot fully judge on this part. The topic of the manuscript is of interest for a broad readership and deserves publication. However, some issues regarding chemical terms the mutational data and the MD simulations have to be addressed before.”

We thank this reviewer for the insightful comments on our manuscript.

“1) The authors should carefully check the manuscript for the right chemical terms/nomenclature. Epinephrine has only one configuration, but can have different conformations, which should not be mixed up (2nd paragraph of the results, Fig. S6, ...). A rotation of a single bond doesn’t result in different stereoisomers (I133-135). The ligand A6160 doesn’t contain an imidazole ring, but an imidazoline ring (e.g. Figure 3, I156, ...).”

As suggested, we have made the wording changes. Thank you!

“2) The mutational studies for understanding the selectivity of A61603 are convincing, but it could benefit from a more detailed discussion. The results for $\alpha 1A$ A189S and $\alpha 1B$ S208A are quite surprising, what’s the explanation for this? The authors just briefly touch this topic. The figure 4C shows the $\alpha 1B$ receptor (pdb entry 7B6W). In this structure the $\alpha 1B$ receptor is bound to an inverse agonist. I wonder whether and how this influences the result. In any case, some modeling of the $\alpha 1B$ receptor would strengthen the manuscript (not necessary to model all mutations!).”

We thank the reviewer for this suggestion. To gain further insight into the effects of individual versus triple mutations, we have performed additional GaMD simulations on an active state structure of α_{1B} -AR. The results of these simulations are in strong agreement with our functional data (see new **Figure S10**). From the simulation studies, α_{1A} -AR(A189S) and α_{1B} -AR(S208A) do not change the structural interactions between A61603 and the receptors. These explain our functional data where α_{1A} -AR(A189S) and α_{1B} -AR(S208A) maintained WT-like properties.

The following texts have been added to the revised manuscript.

We have built a homology model of the active state of α_{1B} -AR using our cryo-EM structure of A61603-bound α_{1A} -AR as the template and conducted GaMD simulations on the wild-type α_{1B} -AR, α_{1B} -AR(S208A), and α_{1B} -AR(A204V,S208A,L314M). In the wild-type α_{1B} -AR and α_{1B} -AR(S208A), A61603 exhibited large RMSDs relative to the starting conformation, with reduced interactions between A61603 and α_{1B} -AR (the new **Figure S10**). In contrast, A61603 in α_{1B} -

AR(A204V,S208A,L314M) exhibited low RMSD at ~ 2.5 Å, forming stable interactions with residues D125^{3,32} and S207^{5,43} in the receptor (the new **Figure S10**).

Figure S10. GaMD simulations of A61603 in complexes with wild-type and mutant α_{1B} -ARs. (a, d, g) Time courses of A61603 RMSDs relative to the initial structures in the system of wild-type α_{1B} -AR (a), α_{1B} -AR(S208A) (d) and α_{1B} -AR(A204V,S208A,L314M) (g). (b, e, h) Time courses of the distance between the CG atom of D125^{3,32} and N2 atom of A61603 in the system of wild-type α_{1B} -AR (b), α_{1B} -AR(S208A) (e) and α_{1B} -AR(A204V,S208A,L314M) (h). (c, f, i) Time courses of the distance between the OG atom of S207^{5,43} and the O2 atom of A61603 in the wild-type of α_{1B} -AR (c), α_{1B} -AR(S208A) (f) and α_{1B} -AR(A204V,S208A,L314M) (i). Three independent 500 ns GaMD simulations are shown for each condition.

“3) Why do the authors use GaMD and not classical MD simulations? I wonder, if calculated free energies would support the findings of the simulated mutants.”

GaMD has been established as a robust technique for unconstrained enhanced sampling and free energy calculations of biomolecules (PMID: 26300708). It works by adding a harmonic boost potential to smooth the potential energy surface and reduce system energy barriers. GaMD has been demonstrated to enhance sampling of various biomolecular interactions and

conformational dynamics, including protein folding, ligand binding, and peptide binding and protein-protein/nucleic acid interactions (PMID: 34899998). Relevant to GPCRs, GaMD simulations have successfully revealed the mechanisms of GPCR activation, ligand binding and GPCR-G protein interactions, which were consistent with experimental data and/or long timescale conventional molecular dynamics (cMD) simulations (PMID: 27791003; PMID: 30442899; PMID: 29507218; PMID: 31283874; PMID: 32515227; PMID: 34497422; PMID: 35835792). Compared with cMD, GaMD has been shown in previous studies to achieve much higher sampling efficiency. In particular, deactivation of an adenosine GPCR upon removal of the G protein and positive allosteric modulator was captured in microsecond simulations using GaMD, but not cMD (PMID: 34497422). Therefore, the GaMD simulations were used in the current study.

“4) I would suggest to reorganize Figure S9 to directly compare systems with the same mutation. The plots show quite some divergent behavior for the single runs. Running simulations in triplicates is state-of-the-art, but if they behave so differently another run (or two) might be necessary. I miss a brief explanation what is happening during the simulations, which might help to address the issue mentioned above.”

As suggested, we have reorganized Figure S9 to directly compare systems with the same mutation.

As pointed out by this reviewer, conducting simulations in triplicates has become a standard practice in the simulation field. In principle, multiple runs are able to enhance sampling, leading to a more comprehensive understanding of the system dynamics. Despite sharing the same initial atomic coordinates, the usage of random seed results in slightly different atomic velocities at the start of GaMD production simulations. Consequently, multiple runs can potentially drive the system towards different low-energy states, thereby increasing the sampling space. By utilizing all the trajectories, we can perform free energy calculations to characterize all the sampled conformational space. We calculated the 2D free energy profiles of the A61603 RMSD and the distance between the CG atom of D106^{3,32} and N2 atom of A61603, as well as the distance between the OG atom of S188^{5,43} and O2 atom of A61603 in both the α_{1A} -AR and α_{1B} -AR systems (the new **Figure S11**). In wild-type and α_{1A} -AR(A189S), one low-energy state of the ligand with a narrow conformational space was sampled. Conversely, the ligand conformational space in the other mutant α_{1A} -ARs were increased with at least two distinct low-energy states. Among the α_{1B} -AR system, α_{1B} -AR(A204V,S208A,L314M) sampled only one low-energy state, while α_{1B} -AR (WT) and α_{1B} -AR(S208A) sampled at least two low-energy states with much larger conformational space (the new **Figure S12**). These findings suggest that the systems with functional responses to A61603 are sampling only a stable A61603 binding pose, whereas the systems without functional responses to A61603 could explore a larger ligand conformational space, indicating reduced stability of A61603. These results are consistent with A61603 RMSF and the MM/GBSA calculations. These observations further support the significance of multiple runs in achieving a more sufficient sampling and comprehensive understanding of the system's behavior.

Figure S11. (a-e) 2D free energy profiles of the agonist A61603 RMSD relative to the distance between the CG atom of D106^{3,32} and N2 atom of A61603 in the system of α_{1A} -AR (WT) (a), α_{1A} -AR(V185A) (b), α_{1A} -AR(A189S) (c), α_{1A} -AR(M292L) (d) and α_{1A} -AR(V185A,A189S,M292L) (e). (f-j) 2D free energy profiles of the agonist A61603 RMSD relative to the distance between the OG atom of S188^{5,43} and O2 atom of A61603 in the system of α_{1A} -AR (WT) (f), α_{1A} -AR(V185A) (g), α_{1A} -AR(A189S) (h), α_{1A} -AR(M292L) (i) and α_{1A} -AR(V185A,A189S,M292L) (j).

Figure S12. (a-c) 2D free energy profiles of the agonist A61603 RMSD relative to the distance between the CG atom of D125^{3,32} and N2 atom of A61603 in the system of α_{1B} -AR (WT) (a), α_{1B} -AR(S208A) (b), α_{1B} -AR(A204V,S208A,L314M) (c). (d-f) 2D free energy profiles of the agonist A61603 RMSD relative to the distance between the OG atom of S207^{5,43} and O2 atom of A61603 in the system of α_{1B} -AR (WT) (d), α_{1B} -AR(S208A) (e), α_{1B} -AR(A204V,S208A,L314M) (f).

Reviewer #2:

“In this study, Su and colleagues have solved two CryoEM structures of the active-state alpha1A adrenergic receptor in complex with a ‘mini-Gq’ transducer and either the endogenous agonist, epinephrine (3 A) or the selective agonist, A61603 (2.6 A). This finding is important because, although alpha adrenergic receptor-targeting therapeutics have been available for many years, they are largely non-selective and thus many opportunities remain for more development of more selective agents for a range of unmet medical needs. By and large, the study is well performed and the application of molecular dynamics and targeted mutagenesis to validate and provide new insights into determinants of agonist selectivity between the related alpha1A and alpha1B subtypes is a useful contribution to the field. I have some comments for consideration by the authors to strengthen the manuscript and their findings.”

We thank this reviewer for the insightful comments on our manuscript.

1. “First, in terms of writing style, I feel that there are a few instances of either ambiguity or over-statement. For instance, in the Introduction, the authors comment on a lack of ‘selective pharmacological agents for alpha1-ARs’ as a general statement, and also cite a relatively old reference (in fact, a number of references are surprising old or not appropriate – more below). Do they mean lack of selective antagonists, selective agonists, or both? Clearly, this paper is focusing on the latter and, as I indicated, there are already many alpha1R-targeting medicines on the market. This needs to be made clearer.”

As suggested, we have changed the word “agents” to “agonists”, and have added more recent references. Thanks!

2. “Second, it is possible that I missed this, but I find it surprising that the authors have not actually characterised the binding affinity of their agonists in their own hands on the constructs used for structural determination. At a minimum, this needs to be done, and it is not sufficient to simply quote ref. 24, Dalal & Grujic, StatPearls (2022). Indeed, I would like to draw the authors’ attention to a recent publication by Proudman and Baker, 2021, Pharmacol. Res. Persp., 9(4): e00799, where they carefully compared the pharmacology of 62 different alpha AR agonists, including A61603, confirming its high selectivity, but speculating that this likely driven by preferential binding rather than preferential signaling efficacy. This is important to include, and relates to the third section of the current MS (“Different configurations of epinephrine....etc.”), where the authors identify interactions critical for ‘...both potency and efficacy’....This statement is incorrect; I assume the authors mean ‘...both affinity and efficacy...’. Thus, in light of the above, it is necessary to have a comparative estimate of the affinity of each of the agonists used, in addition to the functional data presented by the authors.”

As suggested, we have now explicitly stated that the specificity of A61603 is known to be due to affinity differences.

In previous studies, it was demonstrated that the specificity of A61603 for α_{1A} -AR was from the specific binding of A61603 to α_{1A} -AR. This conclusion was confirmed by the mentioned paper (by Proudman and Baker, 2021, our new Ref. 7). Since these publications used similar experimental conditions to ours (for example, human α_{1A} -AR expressed in CHO cells) and we

have validated our findings in terms of both a functional readout and detailed computational studies of binding, we feel it is unnecessary to perform experimental binding studies.

The following text has been added to the revised manuscript.

To further investigate the selectivity of A61603, we performed MM/GBSA binding free energy calculations using GaMD trajectories of all systems. (The MM/GBSA (molecular mechanics energies combined with the Poisson–Boltzmann or generalized Born and surface area continuum solvation methods) are popular approaches to calculate absolute binding affinities.) Due to inherent inaccuracy of entropy calculations, we focused on comparing only the enthalpy values (**Table S2**). For α_{1A} -AR and its mutants, A61603 binding free energy was less favorable, i.e., -22.18 ± 0.73 kcal/mol for α_{1A} -AR(V185A) and -11.52 ± 2.47 kcal/mol for α_{1A} -AR(V185A, A189S, M292L), compared to -26.73 ± 0.22 kcal/mol for the wild-type α_{1A} -AR. α_{1A} -AR(A189S) exhibited a very similar A61603 binding free energy (-26.54 ± 1.89 kcal/mol) as the wild-type α_{1A} -AR. These are consistent with our functional data. For α_{1B} -AR and its mutants, α_{1B} -AR(A204V, S208A, L314M) showed the strongest binding affinity (-28.21 ± 2.80 kcal/mol), followed by α_{1B} -AR(S208A) (-25.81 ± 2.67 kcal/mol) and the wild-type (-11.52 ± 2.47 kcal/mol). Again, these are consistent with our functional data. In general, receptor systems with a stronger functional response to the A61603 binding exhibited more favorable binding free energy compared to those with a weaker or no functional response. However, an exception was observed in α_{1A} -AR (M292L), where the binding free energy was -27.88 ± 1.53 kcal/mol. Possible reasons for this might be the exclusion of entropy and the un-reweighting of the binding free energy due to noise in the calculation.

Table S2. Summary of the A61603 root-mean-square-fluctuation (RMSF) and binding free energy from MM/GBSA calculations on the wild-type and mutant α_{1A} -ARs and α_{1B} -ARs. For each system, 1,000 frames were used for MM/GBSA calculation. Means \pm SD are shown.

α_{1A} -AR					
System	WT	V185A	A189S	M292L	V185A,A189S,M292L
A61603 RMSF (Å)	1.46 \pm 0.10	2.36 \pm 0.52	1.10 \pm 0.09	1.74 \pm 0.39	2.58 \pm 0.33
A61603 binding free energy (kcal/mol)	-26.73 \pm 0.22	-22.18 \pm 0.73	-26.54 \pm 1.89	-27.88 \pm 1.53	-11.52 \pm 2.47
α_{1B} -AR					
System	WT	S208A	A204V,S208A,L314M		
A61603 RMSF (Å)	4.28 \pm 0.30	3.55 \pm 1.97	0.98 \pm 0.21		
A61603 binding free energy (kcal/mol)	-11.52 \pm 2.47	-25.81 \pm 2.67	-28.21 \pm 2.80		

Moreover, we compared A61603's root-mean-square fluctuations (RMSF) among the different models. In this context, larger RMSF values indicate weaker binding of A61603. The calculated RMSF values of A61603 in α_{1A} -AR(WT), α_{1A} -AR(V185A), α_{1A} -AR(A189S), α_{1A} -AR(M292L), and α_{1A} -AR(V185A, A189S, M292L) were 1.46 ± 0.10 Å, 2.36 ± 0.52 Å, 1.10 ± 0.088 Å, 1.74 ± 0.39 Å, and 2.58 ± 0.33 Å, respectively. α_{1A} -AR(A189S) exhibited similar RMSF compared to the wild-type, while the other α_{1A} mutants showed higher A61603 fluctuations. The A61603 RMSF values in α_{1B} -AR(A204V, S208A, L314M), α_{1B} -AR(S208A) and wild-type α_{1B} -AR were 0.98 ± 0.21 Å, 3.55 ± 1.97 Å, and 4.28 ± 0.30 Å, respectively. These are consistent with our functional data.

Additionally, we calculated the 2D free energy profiles of A61603 RMSD and the distance between the CG atom of D106^{3,32} and N2 atom of A61603, as well as the distance between the OG atom of S188^{5,43} and O2 atom of A61603 in both the α_{1A} -AR and α_{1B} -AR systems (the new **Figure S11**). In wild-type α_{1A} -AR and α_{1A} -AR(A189S), one low-energy state of A61603 with a narrow conformational space was sampled. Conversely, the ligand conformational space in the other mutant α_{1A} -ARs were significantly increased with at least two distinct low-energy states. Among the α_{1B} -AR systems, α_{1B} -AR(A204V, S208A, L314M) sampled only one low-energy state, while α_{1B} -AR(WT) and α_{1B} -AR(S208A) sampled at least two low-energy states with much larger conformational space (the new **Figure S12**). These findings suggest that systems with functional response to A61603 are sampling only one stable A61603 binding pose, whereas systems without functional response to A61603 could explore a larger ligand conformational space, indicating reduced stability of A61603. These results are consistent with above A61603 RMSF and MM/GBSA calculations.

All these support previous conclusion that the specificity of A61603 is due to specific binding.

3. *“Third, there are a number of instances where I feel that the authors make too much of the different configurations of epi that can be exploited to produce alpha- or beta-AR selective agonists; either be more explicit about some potential modifications that may be worth pursuing in the future, or else tone this down. The most interesting part of the study in my opinion is the molecular discrimination between a1A and a1b selectivity (many of the early findings on the active states are not particularly surprising when compared to existing active state Class A receptors). The targeted mutagenesis supports that data but, as I note, only in function. I feel that this could also be supported by some carefully selected mutants to validate in binding assays as well.”*

As suggested, we have deleted the two sentences.

4. *“Finally, the Discussion is rather repetitive of the results. I think that, if the data were supplemented with additional binding experiments, it would be more useful for the field to Discuss the implications of affinity-based versus efficacy-based selectivity of such agonists.”*

As suggested, we have shortened the Discussion. (Since the primary focus of the paper is on the structural basis of agonist interactions with α_{1A} -AR, we have not discussed too much on the affinity vs efficacy-based selectivity. As stated in the mentioned paper (by Proudman and Baker, 2021, our new Ref. 7), “little evidence for selective intrinsic efficacy between the compounds, with

perhaps the exception of dobutamine which may have some α_{1D} -selective efficacy". All other compounds showed affinity-based selectivity.)

Reviewers' Comments:

Reviewer #1:

Remarks to the Author:

All the points raised in the first round of review have been adequately addressed and I would like to recommend publication.

Reviewer #2:

Remarks to the Author:

Although the authors did not perform additional binding experiments, which would have conclusively allowed them to compare binding to signaling in a common cellular background, their rationalisation for the assay conditions between their work and other groups that have done such assays, in addition to the new computational experiments in the revision are sufficient to address my main concerns. The conclusion that the observed differences in selectivity are affinity-based (not efficacy-based) should be strongly re-iterated throughout.

List of Manuscript Changes

We thank the reviewers very much for the helpful comments on our manuscript.

Reviewer #1:

“All the points raised in the first round of review have been adequately addressed and I would like to recommend publication.”

We thank this reviewer for the help.

Reviewer #2:

“Although the authors did not perform additional binding experiments, which would have conclusively allowed them to compare binding to signaling in a common cellular background, their rationalisation for the assay conditions between their work and other groups that have done such assays, in addition to the new computational experiments in the revision are sufficient to address my main concerns. The conclusion that the observed differences in selectivity are affinity-based (not efficacy-based) should be strongly re-iterated throughout.”

We thank this reviewer for the help. As suggested, we have added the statement that “the observed differences in selectivity are consistent with a model that is affinity-based, as opposed to efficacy-based” in the results and the discussion.